# Improved Method to Suppress Azimuth Ambiguity for Current Velocity Measurement in Coastal Waters Based on ATI-SAR Systems

**Na Yi** [1] [ID], **Yijun He** [1,2,*] [ID] and **Baochang Liu** [1]

1   School of Marine Sciences, Nanjing University of Information Science and Technology, Nanjing 210044, China; nayi@nuist.edu.cn (N.Y.); bcliu@nuist.edu.cn (B.L.)

2   Laboratory for Regional Oceanography and Numerical Modeling, Qingdao National Laboratory for Marine Science and Technology, Qingdao 266237, China

*   Correspondence: yjhe@nuist.edu.cn

**Abstract:** Measurements of ocean surface currents in coastal waters are crucial for improving our understanding of tidal atlases, as well as for ecosystem and water pollution monitoring. This paper proposes an improved method for estimating the baseline-to-platform speed ratio (BPSR) for improving the current line-of-sight (LOS) velocity measurement accuracy in coastal waters with along-track interferometric synthetic aperture radar (ATI-SAR) based on eigenvalue spectrum entropy (EVSE) analysis. The estimation of BPSR utilizes the spaceborne along-track interferometry and considers the effects of a satellite orbit and an inaccurate baseline responsible for azimuth ambiguity in coastal waters. Unlike the existing methods, which often assume idealized rather than actual operating environments, the proposed approach considers the accuracy of BPSR, which is its key advantage applicable to many, even poorly designed, ATI-SAR systems. This is achieved through an alternate algorithm for the suppression of azimuth ambiguity and BPSR estimation based on an improved analysis of the eigenvalue spectrum entropy, which is an important parameter representing the mixability of unambiguous and ambiguous signals. The improvements include the consideration of a measurement of the heterogeneity of the scene, the corrections of coherence-inferred phase fluctuation (CPF), and the interferogram-derived phase variability (IPV); the last two variables are closely related to the determination of the EVSE threshold. Besides, the BPSR estimation also represents an improvement that has not been achieved in previous work of EVSE analysis. When the improved method is used on the simulated ocean-surface current LOS velocity data obtained from a coastal area, the root-mean-square error is less than 0.05 m/s. The other strengths of the proposed algorithm are adaptability, robustness, and a limited user input requirement. Most importantly, the method can be adopted for practical applications.

**Keywords:** along-track interferometric synthetic aperture radar (ATI-SAR); current line-of-sight (LOS) velocity; coastal waters; azimuth ambiguity; baseline-to-platform speed ratio estimation

## 1. Introduction

Ocean sea surface currents play a key role in air-sea interaction, biological production, and mixing between the upper and lower water layers in coastal areas [1–4]. In addition, their measurement in coastal areas provides important information to fishing and electricity generation industries [5,6].

In coastal waters, tidal currents are one of the most important factors of the sea surface current. Generally, tidal currents are quite deterministic and can also be precisely inferred by in situ measurements. In situ measurement devices, including the acoustic Doppler current profiler

(ADCP) and the current meter, however, have limited coverage and are expensive. On the other hand, the along-track interferometric synthetic aperture radar (ATI-SAR) does not have these limitations; meanwhile, ATI performs well for measurements of the sea surface currents, including the tidal currents [7]. Along-track interferometry (ATI) is a powerful tool for the measurement of ocean currents [8–13]. Interferometry was originally proposed in [14], and is based on processing two interferometric SAR images of the same scene obtained with two antennas within a short time [15,16]. Most of the existing studies on the retrieval of surface currents by interferometric SAR [17–19] assume systems with an accurate baseline and constant platform velocity, i.e., a completely accurate baseline-to-platform speed ratio (BPSR). However, in real-life applications, the baseline is often inaccurate; for example, in the commonly used spaceborne SAR data acquisition mode, and the entire antenna is active in pulse transmission but divided into several parts to receive returns. The effective phase center of each receiving channel is assumed to be located in the middle between the physical transmission and the respective receiving phase center, but this method is not accurate [20]. In addition, the accuracy of BPSR is not considered.

While the airborne ATI [21–24] is usually limited by the achievable coverage and complex logistical requirements, the spaceborne ATI [25–27] can illuminate any point of interest during a certain overpass, and obtain wide-swath and high-resolution real-time current observations [28]. Despite the relatively high degree of azimuth ambiguity, spaceborne InSAR systems perform better in ocean current inversion in open sea. Nevertheless, in coastal waters, azimuth ambiguities may have a negative influence on the accuracy of measurements of the velocity of sea surface currents as spaceborne ATI systems with wide bandwidth are particularly prone to azimuth ambiguity, which can produce a "ghost signature" in images. Azimuth ambiguity is mainly caused by under-sampling of a signal, i.e., the signal received by the radar originates not only from the area of interest but also includes ghost signatures from the surrounding areas. In locations, such as coastal waters, the ghost signals of scatterers with strong backscattered powers on land will be shifted in azimuth and superimposed on a relatively weak signal from the water, as shown in Figure 1. In Figure 1, the InSAR signals are modeled within the Doppler baseband—$PRF/2 \le f_d \le PRF/2$ ($f_d$ is the Doppler frequency, PRF is the pulse repetition frequency). In addition, it has a negative impact on the estimation of the baseline-to-platform speed ratio (BPSR), and consequently, on the accuracy of BPSR-based ocean currents measurements [29]. Also, azimuth ambiguities have a strong influence on the accuracy of measured current line-of-sight (LOS) velocities. In coastal waters [30], it is, therefore, necessary to eliminate azimuth ambiguity before estimating BPSR, which necessitates the development of an improved algorithm that not only suppresses azimuth ambiguity but estimates BPSR as well. The two tasks can conveniently be handled using the Doppler interval (for details, see [31]). The Doppler interval specifically refers to the interval without an azimuth-ambiguity Doppler spectrum, which is the Doppler frequency interval with a starting point and ending point in the mathematical sense. Since the Doppler frequency is linearly dependent on the baseline value, according to the definition of the BPSR, the Doppler frequency and BPSR are also linearly dependent, so the former can be used to estimate the BPSR.

Romeiser et al. [7] proposed suppressing azimuth ambiguity through a pixel-value exclusion operation, which eliminates pixels that have an intensity of less than 10 dB at a certain distance. However, this method cannot work reliably in areas with contrast between the land and water. An alternative approach is spectrum filtering and extrapolation [32], but it reduces the azimuthal resolution. The method of analyzing the eigenvalue spectrum entropy (EVSE) proposed by Liu [31] can automatically estimate a usable range of the Doppler domain and needs only limited user inputs, but assumes an accurate baseline and constant velocity of the platform. However, in practical applications, none of these assumptions is true, which motivated us to improve the method.

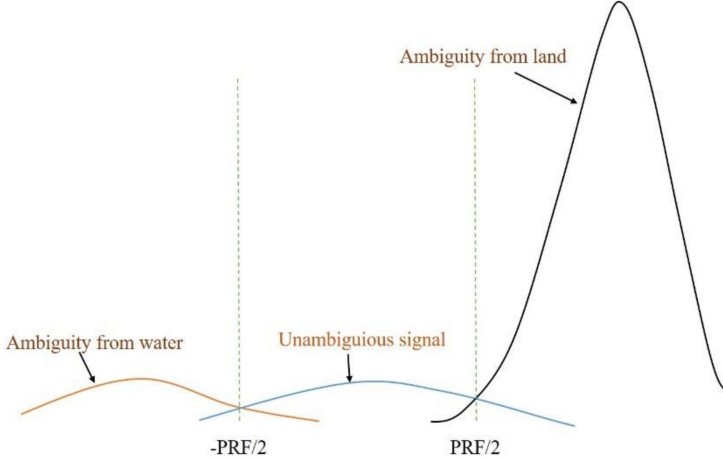

**Figure 1.** Illustration of the Doppler amplitude patterns of the two azimuth ambiguities and the unambiguous signal part. Modified from Liu [31].

This paper proposes an improved algorithm for both azimuth ambiguity suppression and BPSR estimation, considering both the heterogeneity of the scene and BPSR estimation. Although the azimuth ambiguity of spaceborne SAR is relatively high, it has little influence on the inversion of azimuth ambiguity in open ocean regions with uniform scattering. However, azimuth-ambiguity has a great influence on the performance of spaceborne InSAR current measurement in the non-uniform offshore area. There are two main reasons for this. First, the backscattering coefficient of land radar is usually much larger than that of sea radar. Therefore, the azimuth ambiguity component from the land will be superimposed on the sea surface, resulting in a serious decline in the accuracy of InSAR current measurements. Second, the velocity of land ghosting is different from that of sea-surface ghosting, which will also change the measured value of the sea surface current field. If scene heterogeneity is not taken into account, EVSE analysis will fail when applied to practical situations. Furthermore, the improved method can be adopted for practical applications with only limited user inputs. The remainder of this paper is organized as follows. Section 2 describes the proposed method, including an overview of Liu's method [31], an alternative algorithm, and our innovation. Section 3 presents the results of applying the improved method to simulated and measured data. Finally, a discussion is presented in Section 4, and in Section 5, conclusions are drawn.

## 2. Methodology

In this section, to improve the accuracy of current LOS velocity estimation, we develop an alternate algorithm for ambiguity suppression and BPSR estimation based on the method of Liu [31]. The surface velocity corresponds more precisely to a mean motion of scattering elements, and the element velocities are weighted by their normalized radar cross section (NRCS) [33]. Considering that the strong NRCS caused by convergence and divergence of the current can lead to large errors [34], we assumed that the ocean surface was smooth so that we could focus more on the suppression of azimuth ambiguity and BPSR estimation.

An overview of the process of ocean current velocity estimation is shown in the flowchart in Figure 2. The process starts with two original SAR images and ends with the estimation of current velocity. As shown in Figure 2, the flowchart mainly includes three parts: SAR image preprocessing (green rectangle in Figure 2), alternating iteration algorithm (blue rectangle in Figure 2), and velocity estimation (orange rectangle in Figure 2). SAR image preprocessing includes SAR image focusing, the interested area extraction of the area of interest, and conversions from the time domain to the frequency domain via the 2D Fourier transform. An alternating iteration algorithm is the focus of our research, and this algorithm is mainly an alternating iterative algorithm that performs azimuth

ambiguity suppression and BPSR estimation. Finally, we obtain the surface current velocity, which is the LOS velocity.

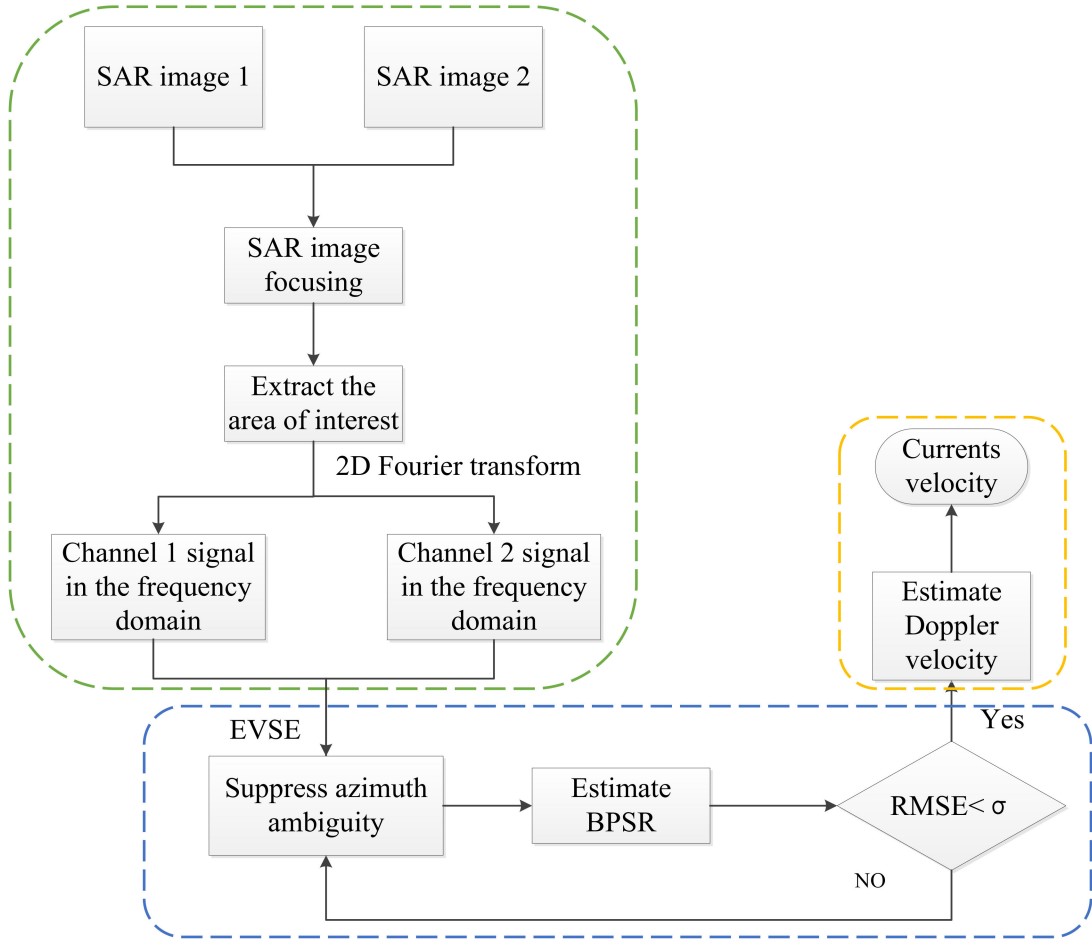

**Figure 2.** Flowchart of the proposed approach.

Selected key procedures underlying the alternate algorithm for ambiguity suppression and BPSR estimation are introduced in this section. The central part of the process includes alternate iterations of ambiguity suppression and BPSR estimation, which is detailed in Figure 3.

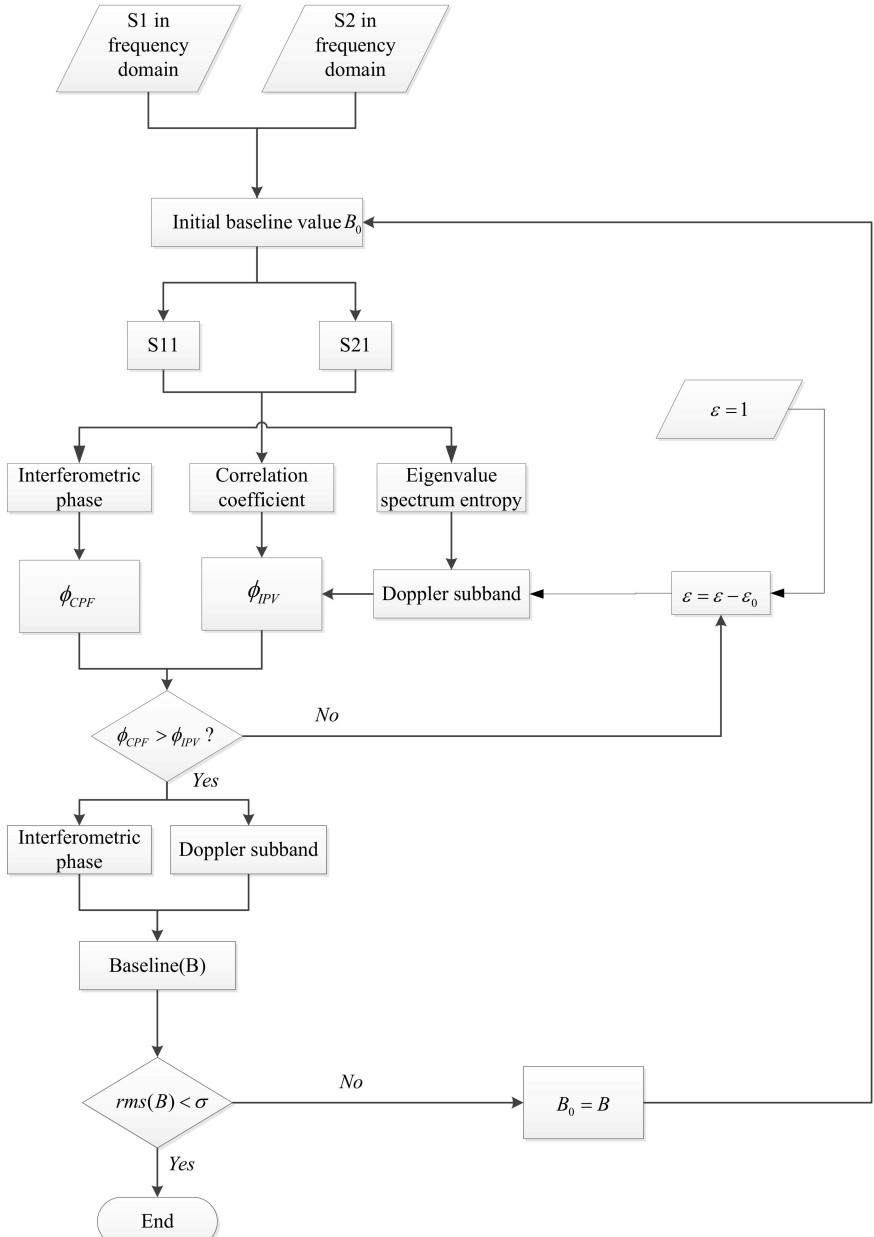

**Figure 3.** Flowchart of the alternate iterative algorithm.

## 2.1. Overview of EVSE Analysis

The method proposed in [31] aims to measure ocean surface currents in coastal waters when the problem of azimuth ambiguity is severe. The velocity estimation is conducted for the sea-surface current in coastal waters. The key component of the method is the analysis of EVSE, which is defined as the entropy of the eigenvalue spectrum of the ATI covariance matrix computed in the Doppler domain. It quantifies the degree of mixing among the Doppler components [31], and is a significant parameter for determining the Doppler domain representation of an unambiguous signal.

The first step of the EVES analysis method is to model the SAR signal. Thus, a dual-channel ATI-SAR signal in the 2-D frequency (the range frequency and the Doppler frequency) domain can be modeled as follows [31]:

$$S_1 = S_{una}^0 + S_{amb}^A + S_{amb}^B + S_{N1} \tag{1}$$

$$
\begin{aligned}
S_2 = \exp&\left\{ j2\pi \cdot \tfrac{B_e}{V_p} \cdot f_d \right\} \\
&\times \begin{bmatrix} \widetilde{S}^0_{una} \cdot \exp\left\{ j\tfrac{4\pi}{\lambda} \cdot \tfrac{B_e}{V_p} \cdot v_r^0 \right\} \\ +\widetilde{S}^A_{amb} \cdot \exp\left\{ j\tfrac{4\pi}{\lambda} \cdot \tfrac{B_e}{V_p} \cdot v_r^A \right\} \times \exp\left\{ +j2\pi \cdot \tfrac{B_e}{V_p} \cdot PRF \right\} \\ +\widetilde{S}^B_{amb} \cdot \exp\left\{ j\tfrac{4\pi}{\lambda} \cdot \tfrac{B_e}{V_p} \cdot v_r^B \right\} \times \exp\left\{ -j2\pi \cdot \tfrac{B_e}{V_p} \cdot PRF \right\} \end{bmatrix} + S_{N2}
\end{aligned}
\tag{2}
$$

where $S_1$ and $S_2$ are the signals received from the two channels of the ATI-SAR system, and $S^0_{una}$ is an unambiguous signal from the fore channel. The parameter PRF represents the pulse repetition frequency. $S^A_{amb}$ and $S^B_{amb}$ are the ambiguous signals from the land and ocean areas, respectively. The tilde-circumflexed signals in $\widetilde{S}^0_{una}$, $\widetilde{S}^A_{amb}$ and $\widetilde{S}^B_{amb}$ are not the same as their uncircumflexed counterparts in Equation (1), because the random motion of the ocean surface affects the received signals. $S_{N1}$ and $S_{N2}$ denote the thermal noise signals of the two channels. $B_e$ is the effective baseline and $V_p$ is the velocity of the radar platform. $f_d$ represents the Doppler frequency and $\lambda$ denotes the radar wavelength.$v_r^0, v_r^A$, and $v_r^B$ are the mean line of sight (LOS) surface velocities of the area of interest, the zone I, and the zone IV, respectively, as shown in Figure 1.

From Equations (1) and (2), the covariance matrix $R$, can be calculated as follows [31]:

$$
\begin{aligned}
R &= E\left\{ \begin{bmatrix} S_1 \\ S_2 \end{bmatrix} \begin{bmatrix} S_1^* & S_2^* \end{bmatrix} \right\} \\
&= \left( P^0_{una} + P^A_{amb} + P^B_{amb} \right) \begin{bmatrix} 1 & \rho \\ \rho^* & 1 \end{bmatrix} + P_n \cdot I_{2\times2}
\end{aligned}
\tag{3}
$$

where $E$ is the expectation operator, $(\cdot)^*$ denotes the complex conjugate operator, $P_n$ is the noise power, and $I_{2\times2}$ is a two-by-two identity matrix. $P^0_{una}$, $P^A_{amb}$, and $P^B_{amb}$ are the powers of the unambiguous signals, the ambiguity of the signal from the land, and the ambiguity of the signal from the ocean, respectively, which can be computed as explained in [31]. Having evaluated the two eigenvalues of the covariance matrix,$R$, denoted as $\lambda_1$ and $\lambda_2$, the EVSE, $H$, of the ATI covariance matrix can be defined as

$$
H = -\left( p_1 \log_2 p_1 + p_2 \log_2 p_2 \right)
\tag{4}
$$

where $p_1$ and $p_2$ are as follows:

$$
p_1 = \frac{\lambda_1}{\lambda_1 + \lambda_2}, p_2 = \frac{\lambda_2}{\lambda_1 + \lambda_2}
\tag{5}
$$

The EVSE quantifies the degree of signal mixing, and is used as the criterion of Doppler domain characterization: the larger the value of EVSE, the higher is the degree of signal component mixing [31].

In Liu's azimuth ambiguity suppression algorithm [31], the EVSE analysis is an important step. As shown in Figure 1, zone II denotes the unambiguous signal. As seen in Figure 4, the fluctuation of the interferometric phase in Doppler frequency is small. Thus, to decide how many Doppler bins should be discarded, a critical EVSE value for the Doppler bins dominated by the unambiguous signal is required. From Liu [31], we can conclude that the determination of zone II depends on three parts: an accurate BPSR, an EVSE curve, and a critical value of EVSE.

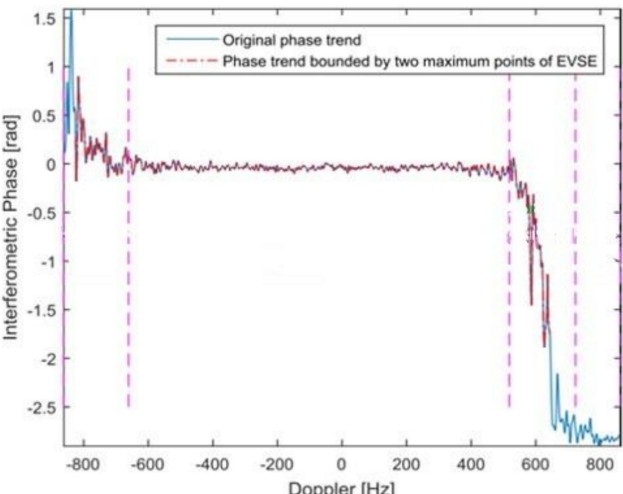

**Figure 4.** Interferometric phase trend after Doppler bin removal based on the two maximum points of the eigenvalue spectrum entropy (EVSE) curve from [31].

As shown in Figure 4, the Doppler bins falling outside the two maximum points of the EVSE curve can be determined by investigating the EVSE curve to find the two maximum points, and the Doppler bins containing an ambiguous signal are excluded. According to Liu [31], the critical value of EVSE is determined such that this critical value identifies a maximum Doppler sub-band over which the following two parameters are equal.

The first parameter, denoted $\phi_{CPF}$, is the coherence-inferred phase fluctuation (CPF), defined as the mean statistical fluctuation of interferometric phases over the Doppler sub-band; and the other parameter, denoted $\phi_{IPV}$, is the interferogram-derived phase variability (IPV), defined as the root-mean-square (RMS) variation of interferogram-derived phase over a certain Doppler sub-band. The expressions for the two parameters are as follows:

$$\phi_{CPF} = \frac{1}{\sqrt{2K}} \frac{\sqrt{1 - \hat{\rho}_M^2}}{\hat{\rho}_M} \tag{6}$$

$$\phi_{IPV} = \sqrt{\frac{1}{L}\sum_{l=1}^{L}\left[\widetilde{\phi}(f_d^{(l)}) - \frac{1}{L}\sum_{p=1}^{L}\widetilde{\phi}(f_d^{(p)})\right]^2} \tag{7}$$

where $K$ is the number of averaged range frequency bins, $\hat{\rho}_M$ is the magnitude of the mean coherence in the 2-D frequency domain, $\widetilde{\phi}(f_d^{(l)})$ is the range-frequency-averaged interferogram phase for the $l$ th Doppler bin $(f_d^{(l)})$ of the Doppler sub-band, and $L$ is the size of the Doppler sub- band. Note that $K$ is based on the assumption that the samples are statistically completely independent and uniform in the range frequency images.

However, when BPSR is not accurate, Liu's method [31] is in effective, which limits the practical applications of the algorithm. In addition, the assumption of $K$ is not correct in practice. The improvements aimed at these two problems in this paper, which will be discussed in the next section, are intended to address this shortcoming to make the method better suited for real-life applications.

## 2.2. Alternate Iteration Algorithm for Azimuth Ambiguity Suppression and BPSR Estimation

The proposed alternate algorithm that can suppress azimuth ambiguity and estimate BPSR, is described below.

The interferometric phase,$\varphi$, and the effective baseline, $B_e$, are related as follows [35]:

$$\varphi = -\frac{2\pi f_d}{V_p} \cdot B_e \tag{8}$$

Based on the above linear relation, the value of the baseline can be obtained from the slope of the phase–frequency curve. Then, BPSR can be shown as

$$BPSR = \frac{B_e}{V_p} \tag{9}$$

As expected, the knowledge of BPSR's accuracy is not sufficient. From Equation (8), we observe that the value of the baseline is related to the Doppler frequency-interference phase. Furthermore, as mentioned in Section 2.1, azimuth ambiguity affects the calculation of the interference phase, and an inaccurate BPSR will result in the failure of the ambiguity suppression algorithm. Therefore, the BPSR and azimuth ambiguity influence each other.

The alternate algorithm for azimuth ambiguity suppression and BPSR estimation are shown in Figure 2, and the detailed flowchart is shown in Figure 3. As the flowchart shows, two adaptive algorithms are executed alternately; one is used to estimate the critical value of EVSE during the process of azimuth ambiguity suppression, and the other is BPSR estimation. There are several key points involved in determining the threshold value of EVSE: first, set $\varepsilon$ as a variable ($0 \leq \varepsilon \leq 1$) with an initial value of 1 in order to determine the characteristic spectral entropy that is less than all of its Doppler units and then combine those Doppler units into a Doppler sub-band; second, calculate CPF ($\phi_{CPF}$) and IPV ($\phi_{IPV}$), when IPV ($\phi_{IPV}$) is larger than CPF ($\phi_{CPF}$), reduce the value of $\varepsilon$ by a certain step size $\varepsilon_0$. Until the condition $\phi_{IPV}(\varepsilon) < \phi_{CPF}$ is established, then the value of $\varepsilon$ is determined as the threshold of EVSE. The Doppler sub-band without ambiguous signal is obtained by discarding all the Doppler units whose EVSE is greater than the EVSE threshold. After removing ambiguity by the EVSE analysis, we obtain the Doppler sub-band that contains the unambiguous signal, from which the baseline value can be estimated using the linear relation between the interferometric phase and the baseline. Next, the baseline value can be used to correct the phase of one of the SAR images, after which the BPSR can be estimated. The process is repeated until the BPSR root-mean-square error is reduced below a predefined small number. It can be seen from Figure 3 that this is also an adaptive algorithm.

*2.3. Correction of IPV and CPF Based on EVSE Analysis*

In the previous sections, an EVSE analysis and an alternate iterative algorithm for the azimuth ambiguity suppression and baseline estimation were discussed. In the current section, we focus on a correction introduced into the method proposed in this paper for non-ideal situations where $K$ in Equation (7) deviates from the original definition in [31].

The correction we added accounts for scene heterogeneity in an SAR image. The heterogeneity of a scene is used to calculate the number of the samples in the range frequency. The so-called effective sample number refers to the number of units of distribution of research objects in an SAR image. In our context, ships and drilling platforms are invalid samples. The sharpness of an SAR image, *shp*, is used to represent the non-uniformity of the scene, and is defined as follows:

$$shp = \frac{\langle I \rangle^2}{\langle I^2 \rangle} = \frac{\left(\frac{1}{L}\sum\limits_{i=1}^{N} I_i^2\right)^2}{\frac{1}{L}\sum\limits_{i=1}^{N} I_i^4} \tag{10}$$

where $I_i$ is the amplitude of the $i^{th}$ pixel in a range compressed image $I$, $< \cdot >$ denotes the spatial average, $N$ is the number of all samples, and $L$ is the number of effective samples.

Therefore, we take *shp* into account is the interferometric phase induced by across baseline. In the proposed algorithm, we take the effect of sharpness of an SAR image into consideration by modifying the *K* in the formula of CPF as follows:

$$\phi'_{CPF} = \frac{\sqrt{1 - \rho_0^2}}{\sqrt{2K \cdot shp \cdot \rho_0}} \tag{11}$$

where $\rho_0$ is the magnitude of the mean coherence, calculated as follows:

$$\rho_0 = \frac{\sum\limits_{i=1}^{L} \left\langle S_{1i} \cdot S_{2i}^* \right\rangle}{\sqrt{\sum\limits_{i=1}^{L} \left\langle S_{1i} \cdot S_{1i}^* \right\rangle \cdot \sum\limits_{i=1}^{L} \left\langle S_{2i} \cdot S_{2i}^* \right\rangle}} \tag{12}$$

where $S_{1i}$ and $S_{2i}$ are the complex values of a corresponding point in $S_1$ and $S_2$, respectively, after $S_2$ has been resampled according to the estimated shift, and *L* is the number of pixels in the sampling area. Note that the numerator is the interferogram while the denominator is the product of the image amplitudes, not powers. The formula for IPV is altered to

$$\phi'_{IPV} = q \cdot \sqrt{\frac{1}{L} \sum_{i=1}^{L} \left[ \widetilde{\phi}(f_d^{(i)}) - \frac{1}{L} \sum_{j=1}^{L} \widetilde{\phi}(f_d^{(j)}) \right]^2} \tag{13}$$

where *q* is a constant used to relax the condition in the computation of IPV. Similarly, the critical value of EVSE is determined such that this critical value identifies a maximum Doppler sub-band over which the above two parameters are equal. Because the BPSR is not accurate in practice, the harsh condition in [31] is also needed to be revised; after several computations, the BPSR tends to be accurate, and *q* will be fixed at 1.

The interferometric phase is computed as

$$\varphi_i = \tan^{-1}\left(\sum_{i=1}^{N} \left\langle S_{1i} \cdot S_{2i}^* \right\rangle \right) \tag{14}$$

where $\varphi_i$ is the $i^{th}$ interferometric phase of the corresponding two SAR images. Note that because a difference of $2\pi$ may be present between the computed and the true interferometric phase, phase unwrapping may be necessary. If there is a $2\pi$ discontinuity in the phase curve, it will cause a large error in the slope of the curve fitted in the Doppler frequency domain and the true interference phase, which will also affect the estimation of BPSR. To alleviate the problem, $\varphi'_i$, can be corrected as follows:

$$\varphi'_i = \varphi_i \pm 2\pi \tag{15}$$

After the above series of corrections, or improvements, the algorithm becomes better suited to practical applications.

In the calculation of surface current LOS velocity, the ocean surface is assumed to be composed of scattering objects that constitute a uniform random surface. The ocean surface current LOS velocity can be computed by

$$V_c = \frac{\sum\limits_{i=1}^{N} \left\langle S_{1i} \cdot S_{2i}^* \right\rangle}{4\pi \cdot BPSR \cdot \sin(\theta)} \cdot \lambda \tag{16}$$

where $\theta$ denotes the incidence angle and $N$ denotes the total number of sample points in the direction of azimuth and range. Because the measured horizontal LOS Doppler velocities are not true current velocities, these measured Doppler velocities for the theoretical contributions of ocean wave motions should be corrected using a numerical model [17].

Liu's method [31] assumes that the value of the effective baseline appearing in Equation (2) is accurate, even though this is often not the case for practical ATI systems. On this basis, we make improvements. Because the algorithm is adaptive, the user only needs to input the SAR data and estimate the platform speed to obtain the BPSR to facilitate the subsequent estimation of the ocean current velocity. In addition, the algorithm is robust and can be applied not only to coastal areas but also land areas, because it estimates the degree of scene heterogeneity. In the following section, the validation data and application results are discussed.

## 3. Results

To assess the feasibility of the improved method, we applied two different sets of data, simulated coastal area data and measured land data. Because of lack of measured coastal data, we used coastal simulation data, which proved to be reliable in [36]. Note that the measured coastal data exist but were not available for this work. Although we do not have the real data from coastal areas, the real data in land area that we have also validates the alternate algorithm. Besides, the real data is also important for validating the scene. The two sets of data represent airborne and spaceborne data, indicating that the proposed algorithm is applicable to both spaceborne and airborne systems. In addition, it also shows that the algorithm is applicable to different scenes such as coastal and land scenes. Both sets of different data are introduced in this section, and the results of azimuth ambiguity suppression and the BPSR estimation processed by the improved algorithm are also shown.

### 3.1. Application to Simulated Data

### 3.1.1. Simulated Data

The simulated raw SAR data of coastal scenes are generated by an inverse omega-k algorithm, whose details can be found in [36] and are not reported here to save space. In the numerical simulation, modulation transfer functions (MTF), including tilt modulation, range modulation, and hydrodynamic modulation, were considered [17]. The simulation parameters were set as in [31], and the key values are listed in Table 1. The range of PRF is about 1000–3000 Hz, and the setting of 1725 Hz is relatively small in this range. However, the selection of PRF is determined by several factors. First, the PRF should satisfy the Nyquist sampling law; second, an excessively large PRF can reduce the unambiguous width and bring range ambiguity; third, PRF selection needs to avoid the echo of sub-satellite point, because this will cause interference in the sampled signal; and lastly, a large PRF comes at the large duty-ratio, which will lead to a large average power and large energy cost. The parameter SNR is the signal-to-noise ratio in ocean surface part and the parameter AASR is the azimuth-ambiguity-to-signal ratio in homogeneous scenes. Note that the effective baseline is 2.4 m and the velocity of the radar platform is 7600 m/s, both of which are closely related to the estimation of BPSR.

**Table 1.** Key simulation parameters for raw SAR (synthetic aperture radar) data of coastal area.

| Parameter | Value |
|---|---|
| PRF (pulse repetition frequency) | 1725 Hz |
| Polarization | VV |
| Radar carrier frequency | 9.6 GHz |
| Effective baseline | 2.4 m |
| Radar platform velocity | 7600 m/s |
| SNR (signal-to-noise ratio) | 6.5 dB |
| Mean water-to-land intensity ratio | −12 dB |
| AASR (azimuth-ambiguity-to-signal ratio) | −20 dB |

The simulation processed SAR image of the coastal area is shown in Figure 5. Figure 5a highlights the azimuth ambiguity, and Figure 5b shows the interferogram phase image. As seen in Figure 5a, the bright objects in the land area produce three ghost signatures in the ocean area. The ghost signatures are also observed in Figure 5b, indicated by the yellow spots. The ghost images observed in both figures demonstrate the necessity to suppress azimuth ambiguity before estimating BPSR by the method introduced in Section 2. The results obtained after removing the ghost images and estimating BPSR are presented in the next section. The interferogram amplitude image of the region marked by the rectangle is shown in Figure 6. This sampling area contains more than 200 pixels.

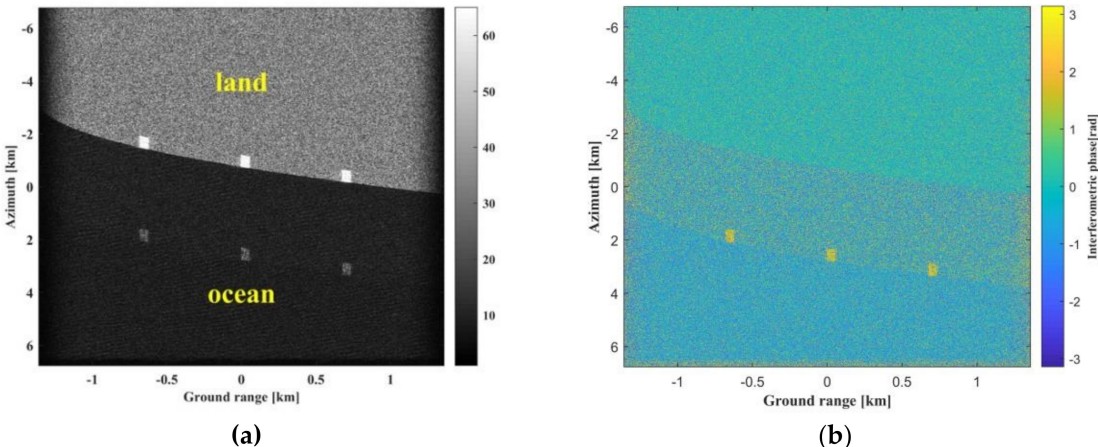

(**a**)                                               (**b**)

**Figure 5.** (**a**) Azimuth ambiguity of the SAR image in the coastal area (note the three bright objects in the land area and their ghost signatures in the ocean area); (**b**) Interferogram phase image.

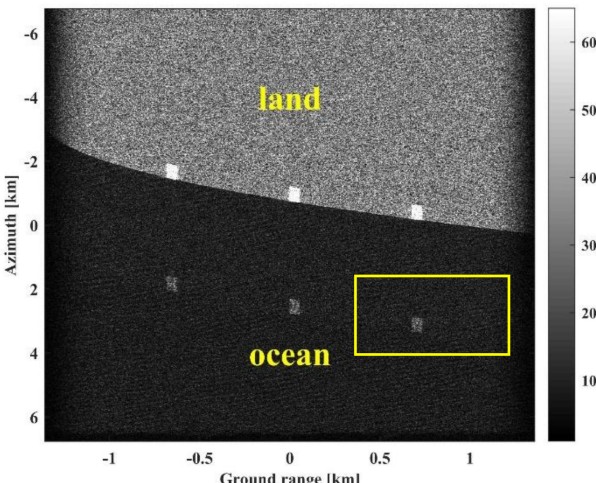

**Figure 6.** Interferogram amplitude image sampled of the region marked by the rectangle.

3.1.2. Results after Processing of the Simulated Data

After processing the data using the alternate iterative algorithm, the Doppler interval in the Doppler spectrum for estimating BPSR is shown in Figure 7, where the red line indicates the starting point of the Doppler range and the blue line indicates the terminal point. As seen in Figure 7, the starting point line is parallel to the terminal point line after four iterations, meaning that the interval tends to be stable between –580 Hz and 460 Hz. The Doppler interval selected by the EVSE analysis is not only used to suppress ambiguity but can also be adopted for estimating the baseline, improving its accuracy, and consequently, the accuracy of the BPSR estimation. From Equation (16), it can be seen that the value of BPSR is inversely proportional to the LOS velocity of the current. That is, when the

BPSR value decreases by $5 \times 10^{-6}$ s, the line-of-sight velocity value will increase by about 0.01 m/s. Therefore, it is necessary to consider the effect of the BPSR value on the LOS velocity. The convergence of the BPSR estimate is shown in Figure 8, where the value of BPSR is found to stabilize at $3.15 \times 10^{-4}$ s after only several iterations. The value of BPSR decreased by 0.17 s compared with the first calculation, so the value of the LOS velocity increased by 0.034 m/s.

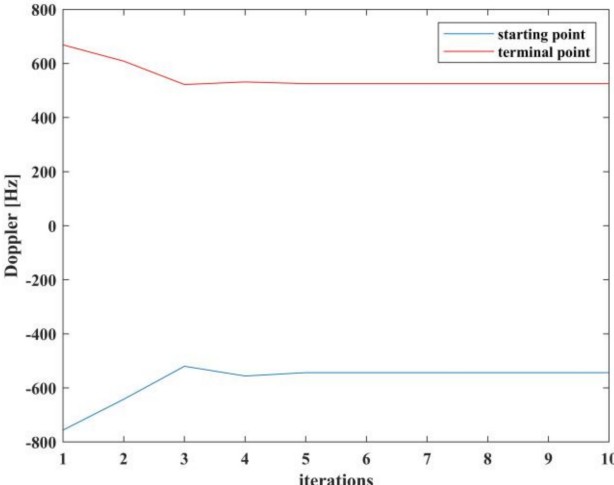

**Figure 7.** Doppler interval endpoint curves after several iterations using the simulated data (the red line indicates the terminal point of the Doppler range, and the blue line indicates the starting point of the Doppler range).

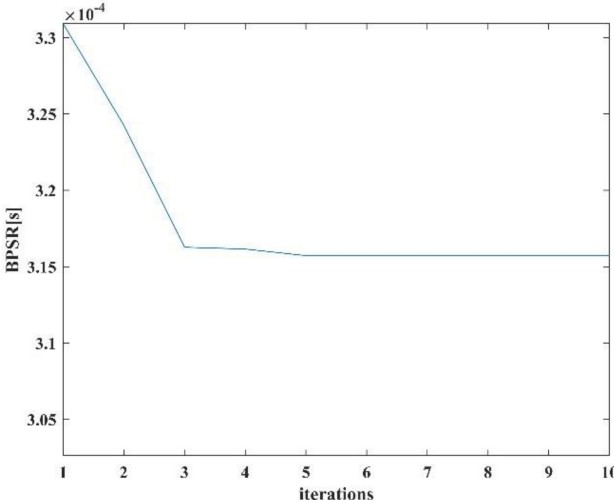

**Figure 8.** BPSR estimation using simulated data and the proposed algorithm.

To obtain a visual impression of the suppression of ghost signatures, we applied the proposed algorithm to the entire ocean surface. The SAR image and the interferogram phase image after the application of the alternate iterative algorithm for azimuth ambiguity suppression and BPSR estimation are presented in Figures 9a and 9b, respectively. Comparing Figure 5a with Figure 9a, and Figure 5b with Figure 9b, it can clearly be seen that the ghost signatures have been removed.

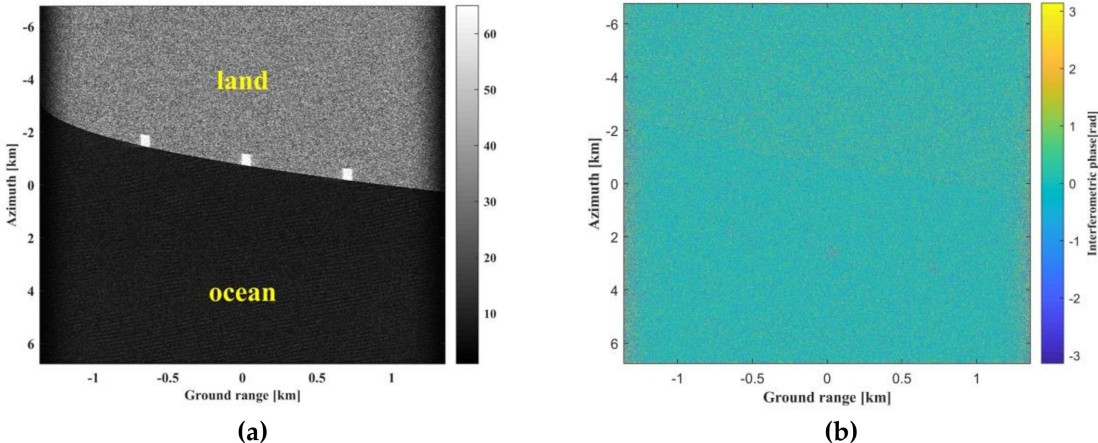

**Figure 9.** (**a**) SAR image after azimuth ambiguity suppression; (**b**) interferogram phase image after azimuth ambiguity suppression.

The Doppler sub-band after azimuth ambiguity suppression is shown as a two phase-frequency curve in Figure 10. Figure 10a corresponds to the first iteration used for selecting the Doppler sub-band, and Figure 10b shows the final iteration. In both figures, the blue line is the original interferometric phase trend, and the red line is that after the azimuth ambiguity suppression. Comparison of Figures 10a and 10b shows that the length of the Doppler sub-band decreases in the iterative process.

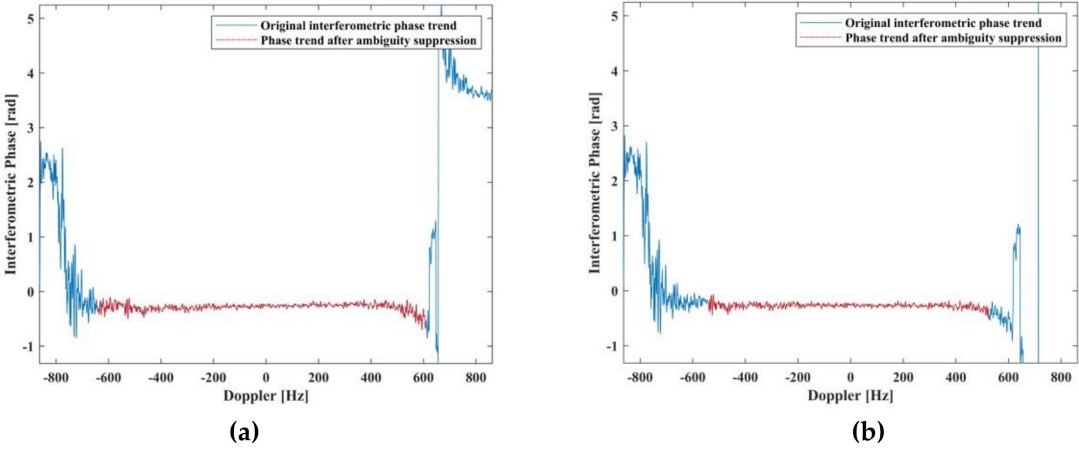

**Figure 10.** (**a**) Phase-frequency curve comparison in the first iteration of ambiguity suppression using simulated data (the blue line is original interferometric phase trend, and the red line is the phase trend after azimuth ambiguity suppression); (**b**) the phase-frequency curve comparison at the final iteration of ambiguity suppression using simulated data.

An estimation of the current velocity was carried out, and the results are shown in Table 2. Assuming a 20% error in BPSR and the true horizontal LOS current velocity of 3.0 m/s, we obtain an estimated mean LOS current velocity of 3.025 m/s, a mean bias of –0.025 m/s, and a standard deviation (STD) of 0.025 m/s. On the other hand, using Liu's method [31], the estimated mean LOS current velocity is 2.543 m/s and the mean bias is 0.457 m/s. There is a larger error in the sea-surface current velocity estimated by Liu's method [31], as highlighted in Table 2. It can thus be concluded that when the BPSR is not accurate, the proposed improved algorithm demonstrates its robustness for the current velocity estimation. Additionally, the results in Table 2 show the improvement of the proposed method compared with the method of Liu [31].

**Table 2.** Current velocity estimates.

| Method | True Horizontal LOS (Line-of-Sight) Current Velocity | Estimated Mean LOS Current Velocity | Mean Bias | STD |
|---|---|---|---|---|
| Liu's method [31] | 3.0 m/s | 2.543 m/s | 0.457 m/s | 0.457 m/s |
| Algorithm proposed in this paper | 3.0 m/s | 3.025 m/s | −0.025 m/s | 0.025 m/s |

The current LOS Doppler velocity maps before and after the application of the improved method are shown in Figure 11a,b, respectively. Both of them are based on the simulation data of true current velocity of 3 m/s and a 20% margin of error in the baseline to calculate the LOS velocity. Figure 11a shows the result without any algorithm, and Figure 11b shows the result obtained by applying the method proposed in this paper. As shown in Figure 11a, affected by azimuth ambiguity, the LOS velocity of the current between the three "ghost" images and shore is about 4 m/s. In the ambiguous areas, the LOS velocity value of the current is further off to –6 m/s. In the open sea (the lower part of the image), the LOS velocity is 5 m/s. Notes that this is not due to ambiguity but is rather due to a 20% error in the baseline. As explained in the introduction, the azimuth ambiguity affects coastal waters but not the open sea. However, in Figure 11b, to obtain a visual impression of the suppression of ghost signatures, the proposed algorithm was applied to the entire ocean surface, and the baseline error and the azimuth ambiguity were both solved based on the application of the improved method, while the LOS velocity is almost the true value (3 m/s). Thus, Figure 11 shows the efficiency of the improved method.

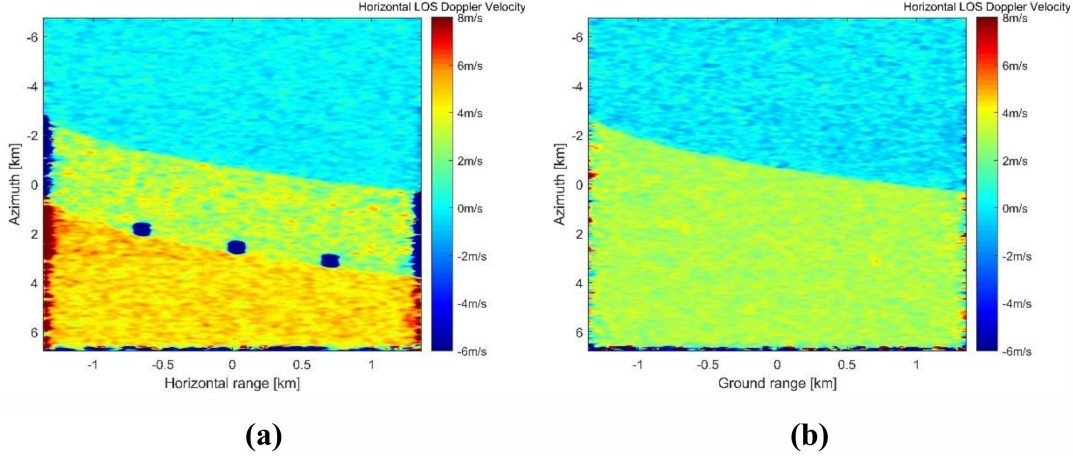

(a)                                                                 (b)

**Figure 11.** Retrieved horizontal LOS (line-of-sight) current Doppler velocity field based on the simulation data with a true current velocity of 3 m/s. (**a**) is without the algorithm application. (**b**) processed with the improved algorithm.

*3.2. Application to Measured Data*

3.2.1. Measured Data

The measured data are acquired over a land area but can nevertheless be processed using the proposed approach. The parameters of the data are listed in Table 3. Again, the two parameters to focus on—effective baseline and radar platform speed—have values of 0.2 m and 110 m/s, respectively. The measured data are unfocused in azimuth, as seen in Figure 12. Figure 12 is a range- compressed azimuth-unfocused SAR image, image-formed for a land area, and the vertical axis is the azimuth direction, while the horizontal axis is the ground range direction. Figure 12 is a piece of the land SAR image.

**Table 3.** Key parameters for the measured data.

| Parameter | Value |
|---|---|
| Wavelength | 0.03 m |
| PRF | 830 Hz |
| Radar carrier frequency | 10 GHz |
| Effective baseline | 0.2 m |
| Radar platform velocity | 110 m/s |
| SNR | 18 dB |
| AASR | −20 dB |

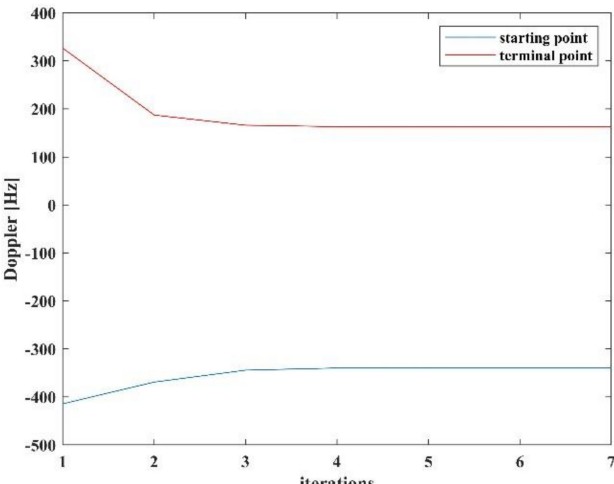

**Figure 12.** Range-compressed azimuth-unfocused SAR image.

### 3.2.2. Results after Processing of Measured Data

The Doppler interval endpoint curves are presented in Figure 13, where the Doppler interval converges quickly. From the estimated BPSR curve in Figure 14, the value of BPSR stabilizes at $1.565 \times 10^{-3}$ s, implying a baseline value of 0.1742 m and a relative error of $1.149 \times 10^{-3}$, respectively.

**Figure 13.** Doppler interval endpoint curve after several iterations using measured data (the red line is the terminal point and the blue line is the starting point of the Doppler range).

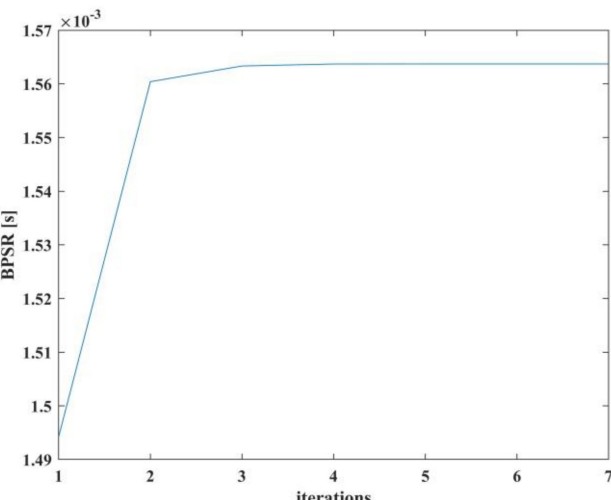

**Figure 14.** BPSR estimate using measured data and the proposed algorithm.

The above analysis demonstrates that although the measured data are from a land area and there is no azimuth ambiguity, BPSR can be estimated using the proposed approach. When the BPSR estimation is added into the method of Liu [31], the algorithm of Liu did not work due to the lack of a specific baseline value. This also shows the improvement of the proposed method.

The result for the case where scene heterogeneity is not considered is shown in Figure 15; the Doppler interval after ambiguity suppression is so narrow that the alternate iterative algorithm cannot be applied, leading to inaccurate BPSR estimation. Besides, unambiguous signals are discarded. However, when scene heterogeneity is taken into account, a Doppler sub-band can be calculated, as shown in Figure 16a, which shows the frequency chosen at the first iteration of the ambiguity suppression procedure. The oscillating parts at both ends of the curve are suppressed in the middle of the Doppler frequency range, as indicated by the red line. Figure 16b illustrates the Doppler sub-band in the last computation of ambiguity suppression. By comparing Figures 16a and 16b, we can conclude that the algorithm performs well, and that it is self-adaptive.

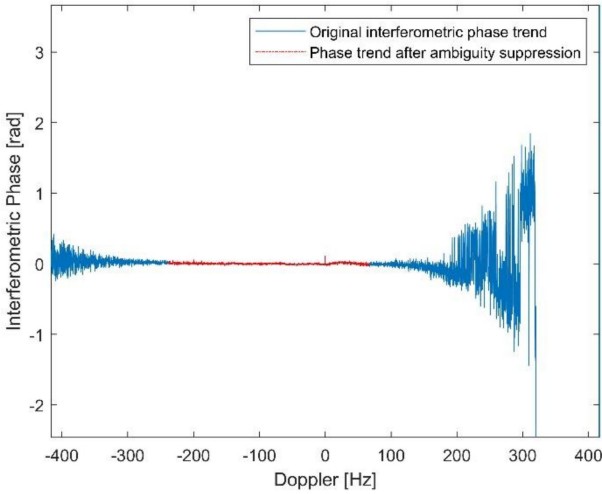

**Figure 15.** Phase-frequency curve comparison without consideration of scene heterogeneity.

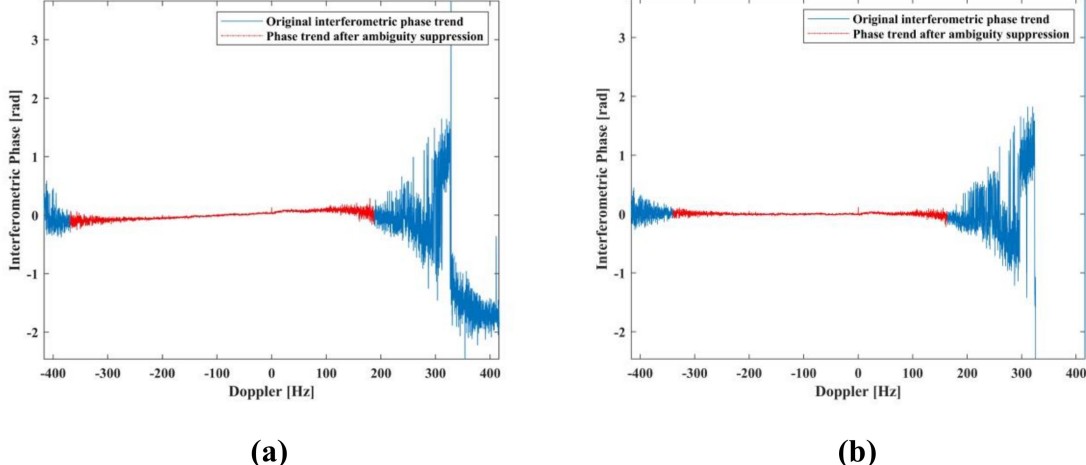

**(a)**                                                    **(b)**

**Figure 16.** Phase-frequency curve comparison in the first iteration of ambiguity suppression using the measured data; (**b**) the phase-frequency curve comparison in the final iteration of ambiguity suppression using the measure data.

## 4. Discussion

It should be noted that the SAR-ATI phase estimates are almost controlled by ocean surface wave motions, which is called the Wind-wave-induced Artifact Surface Velocity (WASV). The wind speed is 5.5 m/s and the current velocity is 0.7 m/s, the WASV reaches 1.6 m/s, which makes a big contribution to the measured ocean surface motion. Mouche et al. [37] provided the first empirical model of the WASV, and the magnitude of the WASV was quantified by Martin et al. [12]. The removal of the contribution from wind-wave is achieved by simulating the SAR Doppler spectra from wind fields proposed by Elyouncha et al. [38]. However, this article does not aim to provide a detailed discussion of separating the current contribution from the wave-induced contribution to the Doppler velocity. Rather, this article focuses on the SAR system and signals related to measurements of ocean surface motion.

The results of azimuth ambiguity suppression and BPSR estimation using the improved alternate iteration algorithm are shown in Section 3. The method of EVSE analysis proposed by Liu [31] is based on an accurate BPSR, which aims at an ideal situation. For a 20% error in BPSR, the current velocity error calculated by Liu's method [31] is larger than that obtained by the improved algorithm in this paper; thus, for the actual situation, the improved algorithm is more effective. This improved algorithm is of great significance for the calculation of sea surface currents in coastal waters. The algorithm is adaptive, as it can be applied not only to spaceborne platforms but also to airborne platforms. Furthermore, the algorithm is also robust as it can be applied to different scenes with different heterogeneity. As shown in Section 3, the simulated data and real data are in different scenes, namely coastal water and land area, respectively. When the baseline is ambiguous or unknown, the improved method can work. Moreover, the TerraSAR-X [7] satellite based on the divided-antenna InSAR mode has strong azimuth ambiguities, and there is also obvious azimuth ambiguity in the ocean SAR image from the GaoFen-3 [39] satellite with ultra-fine strip mode, which seriously affects the data processing of subsequent marine applications. Therefore, the improved algorithm proposed in this paper can not only provide solutions to these problems but also improve the accuracy of the calculation of coastal current velocity. In the proposed algorithm, we did not consider the interference phase caused by the across baseline, which will be investigated and solved in future studies.

## 5. Conclusions

This paper proposes improvements in the algorithm for coastal current velocity measurements that consider real-life, non-ideal conditions and increase the precision of the velocity estimates.

The improved algorithm for the alternate azimuth ambiguity suppression and BPSR estimation can be applied to data from the ATI-SAR systems under relaxed conditions. The proposed approach incorporates a measure of scene heterogeneity, and importantly, is applicable to non-ideal situations with an inaccurate BPSR. The algorithm has successfully been tested on simulated and measured data. Because the measured data from a coastal area were not available, we used simulated data instead and measured data from a land area to test the practicability of the method. Note that data processing has no effect on the separation of wave and sea-surface currents in the subsequent estimation of the sea-surface currents. The processing results of the measured data from the land area also show the importance of considering scene heterogeneity. In addition, the algorithm needs only limited user inputs. After the application of an alternate iterative algorithm for ambiguity suppression and BPSR estimation, the current velocity can be estimated with an error of less than 0.05 m/s. This study indicates that the method can also help to increase the measurement accuracy of the current velocity using both airborne and spaceborne systems, even for systems that have limitations.

**Author Contributions:** Conceptualization: N.Y.; software, N.Y.; validation: N.Y., Y.H. and B.L.; writing-original draft preparation, N.Y.; writing—review and editing: Y.H. and B.L.; supervision: Y.H. All authors have read and agreed to the published version of the manuscript.

**Funding:** This work was supported in part by the National Key Research and Development Program under Grant 2016YFC1401002, the National Natural Science Foundation under Grant 41620104003, and the National Natural Science Foundation under Grant 41606201.

**Conflicts of Interest:** The authors declare no conflict of interest.

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
