# Peer review of "Improved Method to Suppress Azimuth Ambiguity for Current Velocity Measurement in Coastal Waters Based on ATI-SAR Systems"

_remotesensing, doi:10.3390/rs12203288_

Round 1
Reviewer 1 Report
The revised version of the manuscript is more satisfactory than the first version. My comments have been addressed by the authors and the criticisms found in the first version have been largely eliminated.
If the Editor considers this manuscript suitable for Remote Sensing journal, I support its publication.
Specific comment:
Line 257-258: I suggest to clarify here that the measured coastal data exist but are not available for this work (as explained from the authors in the cover letter).
Author Response
Dear Editors and Reviewers:
First, we would like to thank you for all your efforts spent on our manuscript! We would also like to thank the anonymous reviewers for their valuable comments and the suggestions concerning the manuscript! We believe these comments are very useful for improving the quality of this paper.
Those comments are all valuable and very helpful for revising and improving our paper, as well as for providing important guidance, regarding our research. We have studied the comments carefully and have made corrections which we hope might be able to meet with approval, using the "Track Changes" function in Microsoft Word. Revised portions are marked in red and highlighted in the paper. Our English editing certificate is at the end of the comments.
The previous text in the paper is shaded gray, whereas the new text is shaded yellow. The main corrections in the revised paper and our response to the reviewer’s comments are as follows:
Responses to Reviewer #1:
Comments:
- The revised version of the manuscript is more satisfactory than the first version. My comments have been addressed by the authors and the criticisms found in the first version have been largely eliminated.
Response: Thanks very much for the reviewer’s interest in our paper, recognition of our paper, and all the efforts and time spent on our manuscript! We believe the reviewer’s comments and suggestions are very helpful in improving the quality of our paper!
- Line 257-258: I suggest to clarify here that the measured coastal data exist but are not available for this work (as explained from the authors in the cover letter).
Response: We agree with this criticism, and thank you so much. Based on your suggestion, we have added it in pp.10, line 393-394.
Before: Because of the lack of measured coastal data, we used coastal simulation data, which proved to be reliable in [37].
Revised: Because of the lack of measured coastal data, we used coastal simulation data, which proved to be reliable in [37]. Notice that the measured coastal data are not available for this work (although they do exist).
Here is our English editing certificate .(see in PDF)
Special thanks to the reviewer for good comments.
We have no conflicts of interest to disclose.
Please address all correspondence concerning this manuscript to me at yjhe@nuist.edu.cn
Sincerely,
Yijun He

Reviewer 2 Report
This is a revised paper for water current estimation using along-track interferometric synthetic aperture radar (ATI-SAR). The paper seems somehow novel and original while the paper requires extensive improvements.
- The title has no specific information of the "improved method". Anyone can improve something somehow. No one can guess that this paper is related to ATI-SAR and proposes to suppress the azimuth ambiguity.
- In the Abstract, the authors wrote that the azimuth ambiguity suppression is achieved by “an improved analysis of the eigenvalue spectrum entropy” but there is nothing specific information how it is improved. Again, anyone can improve something somehow.
- The descriptions of the methods are insufficient. For example, the descriptions for Eq. (11) are strange. The authors wrote "where K is the number of averaged range frequency bins" in L. 174. On the other hand, "shp" is defined as shp = (|I|^2) / (|I^2|^2) in Eq. 10. This means that sqrt(2K*shp) quickly becomes zero when the value of I is large. (By the way, I is defined as a range compressed image but its size is not specified. Hereafter, this reviewer assumes it is equal to K but please specify in the revised manuscript). This means that the results strongly depend on the size of the window. On the other hand, there is no discussion for the dependency. The authors have to extensively revise the manuscript for clear explanation. Do not expect the readers’ positive understanding.
- Descriptions of formulas are unclear. For example, this reviewer assumes that |S1 S1*| includes averaging or summation but there is no such explanation. It seems as if the authors get only the absolute value.
- The best way to reduce azimuth ambiguity is increasing PRF. From Table 1, the PRF is too small in such a context. The authors have to discuss further. In addition, the authors claimed for the resiliency of the proposal against ambiguous baseline. However, there seems no discussion in the simulation.
- In the experiments section, the authors compared their previous work [32] and the proposal in Table 2 while they compared “before and after the application of the proposed method” in Figure 10. Does this mean Table 2 and Fig. 10 different experiment? If yes, the authors performed unfair comparison. If no, the authors have to avoid misleading descriptions.
- As long as this reviewer sees Figure10, the average horizontal Doppler velocity does not seem approx. 3m/s (dark blue) but slower (light blue). Is the color scale correct? Where is the sampled region? It is strange that 8m/s and -6m/s are plotted in the same color.
- In Section 3.2 the authors performed experiments only with the proposal but have to compare the results as same as in Section 3.1 to show the improvements in the real data.
Author Response
Dear Editors and Reviewers:
First, we would like to thank you for all your efforts spent on our manuscript! We would also like to thank the anonymous reviewers for their valuable comments and the suggestions concerning the manuscript! We believe these comments are very useful for improving the quality of this paper.
Those comments are all valuable and very helpful for revising and improving our paper, as well as for providing important guidance, regarding our research. We have studied the comments carefully and have made corrections which we hope might be able to meet with approval, using the "Track Changes" function in Microsoft Word. Revised portions are marked in red and highlighted in the paper. Our English editing certificate is at the end of the comments.
Our figures are displayed in the PDF.
The previous text in the paper is shaded gray, whereas the new text is shaded yellow. The main corrections in the revised paper and our response to the reviewer’s comments are as follows:
Responses to Reviewer #2:
Comments:
- The paper seems somehow novel and original while the paper requires extensive improvements.
Response: We would like to express our sincere gratitude to the reviewer for his/her efforts and time spent on our manuscript as well as for so detailed remarks and suggestions on our manuscript!
- The title has no specific information of the "improved method". Anyone can improve something somehow. No one can guess that this paper is related to ATI-SAR and proposes to suppress the azimuth ambiguity.
Response: We are sorry that we did not express it clearly in the title, and we agree with this criticism. Thus, we revised it.
Before: Improved Method for Current Velocity Measurement in Coastal Waters
Revised: Improved Method to Suppress Azimuth Ambiguity for Current Velocity Measurement in Coastal Waters Based on ATI-SAR Systems.
- In the Abstract, the authors wrote that the azimuth ambiguity suppression is achieved by “an improved analysis of the eigenvalue spectrum entropy” but there is nothing specific information how it is improved. Again, anyone can improve something somehow.
Response: Yes, we agree with you, and we are sorry that we did not illustrate our improvements specifically. And there is a mistake in the expression “an improved analysis of the eigenvalue spectrum entropy”, in fact, “an improved method based on analysis of the eigenvalue spectrum entropy”. Based on your helpful suggestion, we add the illustrations about the improvements in the Abstract.
Before: This is achieved through an alternate algorithm for azimuth ambiguity suppression and BPSR estimation based on an improved analysis of the eigenvalue spectrum entropy, which is an important parameter representing the mixability of unambiguous and ambiguous signals.
Revised: This is achieved through an alternate algorithm for the suppression of azimuth ambiguity and BPSR estimation based on an improved analysis of the eigenvalue spectrum entropy, which is an important parameter representing the mixability of unambiguous and ambiguous signals. The improvements include the consideration of a measurement of the heterogeneity of the scene, the corrections of coherence-inferred phase fluctuation (CPF), and the interferogram-derived phase variability (IPV), and the last two variables are closely related to the determination of the EVSE threshold. Besides, the BPSR estimation also represents an improvement that has not been achieved in previous work of EVSE analysis. (pp.1, line 18-26)
- The descriptions of the methods are insufficient.
(1)For example, the descriptions for Eq. (11) are strange. The authors wrote "where K is the number of averaged range frequency bins" in L. 174. On the other hand, "shp" is defined as shp = (|I|^2) / (|I^2|^2) in Eq. 10. This means that sqrt(2K*shp) quickly becomes zero when the value of I is large.
(2) (By the way, I is defined as a range compressed image but its size is not specified. Hereafter, this reviewer assumes it is equal to K but please specify in the revised manuscript). This means that the results strongly depend on the size of the window.
(3)On the other hand, there is no discussion for the dependency. The authors have to extensively revise the manuscript for clear explanation. Do not expect the readers’ positive understanding.
Response: We agree that the descriptions of the methods are insufficient.
(1) Thanks for raising a question about Eq. (10). After carefully checking out this formula and the corresponding results, we found that the expression for “shp” used in our program codes and the one presented in the text of our manuscript are different. As a matter of fact, the formula involved in our codes is indeed correct, but unfortunately, we failed to present it correctly in our manuscript. This comment is very important for the paper, and thank you so much! Besides, it included a summation concept, so we revised this formula. (pp.9, line 322-324)
Before: The sharpness of an SAR image, , is used to represent the non-uniformity of the scene, and is defined as follows:
(10)
where is a range compressed image.
Revised: The sharpness of an SAR image, , is used to represent the non-uniformity of the scene, and is defined as follows:
(10)
where is the amplitude of the ith pixel in a range compressed image I, < · > denotes the spatial average, N is the number of all samples, and L is the number of effective samples.
(2) I is defined as a range compressed image and its size is the number of effective samples, L, and it is equal to K. At present, we have some difficulties in deriving an exact analytical formula characterizing the wind size beyond which a reliable estimate of the non-uniformity of the scene, shp, can be obtained. However, as a general rule, dozens of samples contained in the window are usually enough to obtain a reliable estimate of shp. Please note that, in our work, the wind size used to calculate “shp” is more than two hundred, which we believe is sufficient to guarantee an accurate estimate of the parameter, “shp”.
(3) However, in general, the window containing dozens of samples can be enough to process in the algorithm, in terms of our work, we used a window containing more than two hundred samples.
In addition, to make the description of the method section clearer, we have made the following changes:
Before: As shown in Figure 2, the flowchart mainly includes three parts: SAR image preprocessing (green rectangle in Figure 2), alternating iteration algorithm (blue rectangle in Figure 2) and velocity estimation (orange rectangle in Figure 2).
Revised: As shown in Figure 2, the flowchart mainly includes three parts: SAR image preprocessing (green rectangle in Figure 2), alternating iteration algorithm (blue rectangle in Figure 2) and velocity estimation (orange rectangle in Figure 2). SAR image preprocessing includes SAR image focusing, the interested area extraction of the area of interest, and conversions from the time domain to the frequency domain via the 2D Fourier transform. An alternating iteration algorithm is the focus of our research, and this algorithm is mainly an alternating iterative algorithm that performs azimuth ambiguity suppression and BPSR estimation. At last, we get the surface current velocity, which is the LOS velocity. (pp.3, line 135-140)
Before: The alternate algorithm for azimuth ambiguity suppression and BPSR estimation is shown in Figure 2, and the detailed flowchart is shown in Figure 4. After removing ambiguity by the EVSE analysis, we obtain the Doppler sub-band that contains the unambiguous signal, from which the baseline value can be estimated using the linear relation between the interferometric phase and the baseline. Next, the baseline value can be used to correct the phase of one of the SAR images, after which the BPSR can be estimated. The process is repeated until the BPSR root-mean-square error reduces below a predefined small number. It can be seen from 错误! 未找到引用源。 that this is also an adaptive algorithm.
Revised: The alternate algorithm for azimuth ambiguity suppression and BPSR estimation is shown in Figure 2, and the detailed flowchart is shown in Figure 4. As the flowchart shows, two adaptive algorithms are executed alternately; one is used to estimate the critical value of EVSE during the process of azimuth ambiguity suppression, and the other is BPSR estimation. There are several key points involved in determining the threshold value of EVSE: first, set as a variable () with an initial value of 1 in order to determine the characteristic spectral entropy that is less than all of its Doppler units and then to combine those Doppler units into a Doppler subband; second, calculate CPF () and IPV (), when IPV () is larger than CPF (), reduce the value of by a certain step size . Until the condition is established, then the value of is determined as the threshold of EVSE. The Doppler subband without ambiguous signal is obtained by discarding all the Doppler units whose EVSE is greater than the EVSE threshold. (pp.7, line 274-283)
- Descriptions of formulas are unclear. For example, this reviewer assumes that |S1 S1*| includes averaging or summation but there is no such explanation. It seems as if the authors get only the absolute value.
Response: Thanks for this criticism. |S1 S1*| does includes the meaning at the level of mathematical expectation, the expression should have a summation symbol, and we had bot explanation for it. Based on your criticism, we revised formula (12).
Before: where is the magnitude of the mean coherence calculated as follows:
(12)
Revised: where is the magnitude of the mean coherence calculated as follows:
(12)
where S1i and S2i are the complex values of a corresponding point in S1 and S2, respectively, after S2 has been resampled according to the estimated shift, and L is the number of pixels in the sampling area. Note that the numerator is the interferogram while the denominator is the product of the image amplitudes, not powers. (pp.9, line 330-334)
In addition to the modification of formula (12), we also modified formula (13), (14) and formula (16),and added the explanation of the formula as follows:
Before: The formula for IPV is altered to
(13)
where is a constant used to relax the condition in computation of IPV.
Revised: The formula for IPV is altered to
(13)
where is a constant used to relax the condition in the computation of IPV. Similarly, the critical value of EVSE is determined such that this critical value identifies a maximum Doppler subband over which the above two parameters are equal. (pp.9, line 330-334)
Before: The interferometric phase is computed as
(14)
where is the interferometric phase of the two SAR images.
Revised: The interferometric phase is computed as
(14)
where is the interferometric phase of the corresponding two SAR images. (pp.9, line 342-343)
Before: The ocean current velocity can be computed by
(16)
where denotes the incidence angle.
Revised: The ocean surface current LOS velocity can be computed by
(16)
where denotes the incidence angle, N denotes the total number of all the sample points in the direction of azimuth and range. (pp.10, line 381-383)
- (1) The best way to reduce azimuth ambiguity is increasing PRF. From Table 1, the PRF is too small in such a context. The authors have to discuss further.
(2) In addition, the authors claimed for the resiliency of the proposal against ambiguous baseline. However, there seems no discussion in the simulation.
Response: (1) Thanks for your suggestions about the PRF. Yes, we agree that if PRF is large enough, for example, two times as high as it was before, 3450Hz, the raw data are almost unambiguous in azimuth in their original form. However, the PRF we set in simulated data is based on some factors as follows:
- The PRF should satisfy Nyquist sampling law, which make the SAR data can be imaged.
- A too large PRF can reduce the unambiguous width, as a result, it will bring range ambiguity into SAR images.
- The PRF is chosen in a range because the radar cannot receive or transmit echoes when transmitting or receiving them, which restricts the PRF.
- The PRF selection needs to avoid the echo of Nadir, because the echo of Nadir will cause interference to the sampled signal, and sometimes the interference can reach 20dB.
- A large PRF comes at the large duty-ratio, which will make a large average power and big energy cost.
Therefore, we think the PRF set by our simulation data is reasonable. We have also added a description of the PRF setup in the text.
Before: The simulation parameters are adopted by referring to [32], and the key values are listed in Table 1.
Revised: The simulation parameters were set as in [32], and the key values are listed in Table 1. The range of PRF is about 1000-3000Hz, and the setting of 1725 Hz is relatively small in this range. However, the selection of PRF is determined by several factors. First, the PRF should satisfy the Nyquist sampling law; second, an excessively large PRF can reduce the unambiguous width and bring range ambiguity; third, PRF selection needs to avoid the echo of sub-satellite point, because this will cause interference in the sampled signal; and lastly, a large PRF comes at the large duty-ratio, which will lead to a large average power and large energy cost. ( pp.10, line 410-416)
(2) We are sorry that we had not explained clearly enough about the resiliency of the proposal against ambiguous baseline, and thanks a lot. The resiliency includes adaptability and robustness as we claimed in the paper. The improved method can be applied not only to spaceborne platforms, but also to the airborne platforms. The robustness of the proposal against ambiguous baseline we discussed is mainly embodied in applications to the different scenes with different heterogeneity. For example, the simulated data and real data are in different scenes, one is coastal water, the other is land area. When the baseline is ambiguous or not known, the improved method can work. Based on your helpful suggestion, we discuss more in the revision.
Before: This improved algorithm is of great significance for the calculation of sea surface currents in coastal waters.
Revised: The algorithm is adaptive, as it can be applied not only to spaceborne platforms but also to the airborne platforms. Furthermore, the algorithm is also robust as it can be applied to different scenes with different heterogeneity. As shown in Section 3, the simulated data and real data are in different scenes, namely coastal water and land area, respectively. When the baseline is ambiguous or unknown, the improved method can work. (pp.18, line 756-761)
- In the experiments section, the authors compared their previous work [32] and the proposal in Table 2 while they compared “before and after the application of the proposed method” in Figure 10. Does this mean Table 2 and Fig. 10 different experiment? If yes, the authors performed unfair comparison. If no, the authors have to avoid misleading descriptions.
Response: We agree with this criticism. We had not expressed this clearly. Table 2 and Fig. 10 are different experiment. In Table 2, we compared our improved algorithm and previous work [32], the data applied to these two methods based on 20% error in BPSR and the true horizontal LOS current velocity of 3.0 m/s. This result aims at presenting the improvement. However, in Figure 10, whose purpose is to assess the efficiency of our improved method, the comparison is only in the improved method proposed in this paper, and Figure 10 shows the different results, (a) is the result without any algorithm application, and (b) is the result with our improved algorithm. Based on your criticism, we revise our expression as follows: first, we separate these two comparisons in two paragraphs in forms, then we add some explanations into this paper.
Before: The current line-of-sight (LOS) velocity maps before and after the application of the proposed method are shown in Figure 10 (a) and (b), respectively. Both of them are based on the simulation data of true current velocity of 3m/s and a 20% margin of error in baseline to calculate the LOS velocity. As Figure 10(a) shows, affected by azimuth ambiguity, the LOS velocity of the current between the three “ghost” images and shore is about 4m/s. In the ambiguous areas, the LOS velocity value of the current is further off to 8m/s. In the open sea (the lower part of the image), the LOS velocity is 5 m/s. Notes that, this is not due to ambiguity, but due to a 20% error in the baseline. As explained in the introduction, the azimuth ambiguity affects coastal waters but not the open seas. However, in Figure 10 (b), the baseline error and the azimuth ambiguity are both solved based on the application of the method, and its LOS velocity is almost the true value (3m/s).
- Current velocity estimates.
|
Method |
True horizontal LOS current velocity |
Estimated mean LOS current velocity |
Mean bias |
STD |
|
Liu’s method [32] |
3.0 m/s |
2.543 m/s |
0.457 m/s |
0.457 m/s |
|
Algorithm proposed in this paper |
3.0 m/s |
3.025 m/s |
-0.025 m/s |
0.025 m/s |
Revised: Additionally, the results in Table 2 show the improvement of the proposed method compared with the method of Liu [32].
- Current velocity estimates.
|
Method |
True horizontal LOS current velocity |
Estimated mean LOS current velocity |
Mean bias |
STD |
|
Liu’s method [32] |
3.0 m/s |
2.543 m/s |
0.457 m/s |
0.457 m/s |
|
Algorithm proposed in this paper |
3.0 m/s |
3.025 m/s |
-0.025 m/s |
0.025 m/s |
The current LOS Doppler velocity maps before and after the application of the improved method are shown in Figure 11 (a) and (b), respectively. Both of them are based on the simulation data of true current velocity of 3 m/s and a 20% margin of error in the baseline to calculate the LOS velocity. In Figure 11 (a) shows the result without any algorithm, and Figure 11 (b) shows the result obtained by applying the method proposed in this paper. As shown in Figure 11(a), affected by azimuth ambiguity, the LOS velocity of the current between the three “ghost” images and shore is about 4 m/s. In the ambiguous areas, the LOS velocity value of the current is further off to – 6 m/s. In the open sea (the lower part of the image), the LOS velocity is 5 m/s. Notes that this is not due to ambiguity but is rather due to a 20% error in the baseline. As explained in the introduction, the azimuth ambiguity affects coastal waters but not the open sea. However, in Figure 11 (b), to obtain a visual impression of the suppression of ghost signatures, the proposed algorithm was applied to the entire ocean surface, and the baseline error and the azimuth ambiguity were both solved based on the application of the improved method, and the LOS velocity is almost the true value (3 m/s). Thus, Figure 11 shows the efficiency of the improved method. (pp.14-15, line 592-640)
- As long as this reviewer sees Figure10, the average horizontal Doppler velocity does not seem approx. 3m/s (dark blue) but slower (light blue). Is the color scale correct? Where is the sampled region? It is strange that 8m/s and -6m/s are plotted in the same color.
Response: Thanks for this helpful criticism. In Figure 10, the average horizontal Doppler velocity is about 3m/s, but in the Figure, the color presented may be a little wrong. The color scale is correct, but its presentation does confused readers. The 8m/s and -6m/s should have been presented in different, but they may be closed in color. It is a problem that may mislead readers. So we make a revision about the color scale. Based on your concern, the revised Figure 10 can be presented better and will not misguide readers, so thank you again. Here, the revision become Figure 11, because we add a Figure 6. To get a visual impression of the suppression of ghost signatures, we have applied the proposed algorithm to the entire ocean surface. In addition, we add a Figure of the sampled area as Figure 6 shows. ( pp.15,line 644-648; pp.12, line 482-483)
Before:
- (b)
- Retrieved horizontal LOS current field based on the simulation data of true current velocity of 3m/s. (a) is without the algorithm application. (b) is processed by the improved algorithm.
Revised:
- (b)
- Retrieved horizontal LOS current Doppler velocity field based on the simulation data with a true current velocity of 3 m/s. (a) is without the algorithm application. (b) processed with the improved algorithm.
The interferogram amplitude image sampled of the region marked by the rectangle, and it is shown in Figure 6. This sampling area contains more than 200 pixels.
.
- Interferogram amplitude image sampled of the region marked by the rectangle.
- In Section 3.2 the authors performed experiments only with the proposal but have to compare the results as same as in Section 3.1 to show the improvements in the real data.
Response: Yes, it should have been compared the results in Section3.2. Thanks very much. However, the real data is not the same as the simulated data, because it is the real data of land area, not coastal waters, and the horizontal LOS current Doppler velocity can not be gained. In addition, even though the effect of ambiguity suppression cannot be seen in the imaging of the measured data, the Doppler interval also can be gain, and the BPSR can be acquired as well when the effective baseline is not known. Certainly, we used the previous method [32] and added the BPSR estimation; as a result, it did not work, because of no specific baseline value. Based on your suggestion, we add the explanation in the text.
Before: The above analysis demonstrates that although the measured data are from a land area and there is no azimuth ambiguity, BPSR can be estimated using the proposed approach.
Revised: The above analysis demonstrates that although the measured data are from a land area and there is no azimuth ambiguity, BPSR can be estimated using the proposed approach. When the BPSR estimation is added into the method of Liu [32], the algorithm of Liu did not work due to the lack of a specific baseline value. This also shows the improvement of the proposed method. (pp.17, line 709-711)
Here is our English editing certificate.
Special thanks to the reviewer for good comments.
We have no conflicts of interest to disclose.
Please address all correspondence concerning this manuscript to me at yjhe@nuist.edu.cn
Sincerely,
Yijun He

Round 2
Reviewer 2 Report
The paper seems fine to be published.
This manuscript is a resubmission of an earlier submission. The following is a list of the peer review reports and author responses from that submission.
Round 1
Reviewer 1 Report
Row 52: "...the ghost signals of scatterers with strong backscattered powers on land will be shifted in azimuth ..."
The azimuthal shift of the SAR image takes place only if the scatterer is moving. The land itself is not moving (if the authors do not imply cars and trains).
Fig1. What does it mean fPRF?
Reviewer 2 Report
Key idea is to help reduce the impact of ambiguous signals to retrieve SAR-ATI phase information. The baseline-to-platform speed can indeed be largely impacted over mixed-pixel area, e.g. coastal land-sea transition, to lower the capability to retrieve ocean surface current.
First, in the introduction, it must be clarified that estimates are solely concerned by measurements in the line-of-sight direction. It should also be mentioned that tidal currents are quite deterministic, in both space and time, and that in situ (point) measurements, if existing, are generally sufficient to precisely infer tidal currents. Questions may thus be related to local bathymetry effects, and/or wind events. Needless to say that satellite sampling (low revisit time) over coastal region will not accommodate to rapidly varying cases of interests (e.g. wind events, particular wave transformation over bathymetry).
The second aspect, technical, key to help interpret the measurements, is to also consider the initial Doppler centroid (single antenna phase estimate) and Normalized Radar Cross Section (NRCS). Strong NRCS gradient can lead to large errors, and must be detected before applying algorithms to precisely retrieve the SAR-ATI phase shifts. Thus, the proposed flowchart algorithm may not appear optimal.
Finally, the numerical demonstration does not take into account the fact that, to first and large order, SAR-ATI phase estimates are controlled by ocean surface wave motions. Over coastal region, surface wave field will rapidly evolve to further introduce varying biases, to impact the ocean surface current estimates.
Reviewer 3 Report
General Comments
This paper aims to propose a new method for the estimation of the baseline-to-platform speed ratio in order to improve the current measurement accuracy obtained with ATI-SAR in coastal waters.
I am not convinced that this paper is really suitable for the publication in Remote Sensing because it is actually a methodological work and there is no application to real data that clearly shows how the accuracy of current data is increased. The authors themselves apply their method to simulated SAR data, saying that there is no real data to apply it to. So my question is: if there is no such data, what is this method for? Who or what is it useful for?
In general, what is represented in the figures is not accurately described in the text, and this makes the work unclear to the reader. I believe that the work lacks a more detailed and simple introduction useful to focus on the problem and to understand the importance of the various parameters. Some concepts (e.g. Doppler interval for starting and ending points, scene heterogeneity,…etc.) are very nebulous because their importance and usefulness are not clearly explained by the authors.
I would expect to see a map of a current field before and after the application of this method to really understand how the current field can be improved.
Specific Comments:
Figure 1: Explain more accurately in the text what this figure means/represent (What are the different Zone? What do they represent in reality?). Explain the acronym PRF;
Eq. 2, Table 1, Table 3: Give an explanation of the parameters PRF, SNR, AASR;
Line 248: Starting from this line check the references to figures in the text;
Figure 10 is impossible to understand for me.